# On-Site Evaluation of Constituent Content and Functionality of *Perilla frutescens* var. *crispa* Using Fluorescence Spectra

**DOI:** 10.3390/molecules28207199

**Published:** 2023-10-20

**Authors:** Hidemichi Sano, Satoru Kawaguchi, Toshifumi Iimori, Masahiro Kuragano, Kiyotaka Tokuraku, Koji Uwai

**Affiliations:** Graduate School of Engineering, Muroran Institute of Technology, 27-1 Mizumoto-cho, Muroran 050-8585, Japan; 22041042@mmm.muroran-it.ac.jp (H.S.); skawaguchi@mmm.muroran-it.ac.jp (S.K.); gano@mmm.muroran-it.ac.jp (M.K.); tokuraku@mmm.muroran-it.ac.jp (K.T.)

**Keywords:** Aβ aggregation inhibitory activity, antioxidation, difference spectral index (DSI), nondestructive analysis, normalized difference spectral index (NDSI), polyphenols, ratio spectral index (RSI)

## Abstract

*Perilla frutescens* leaves are hypothesized to possess antioxidant and amyloid-β (Aβ) aggregation inhibitory properties primarily due to their polyphenol-type compounds. While these bioactivities fluctuate daily, the traditional methods for quantifying constituent contents and functional properties are both laborious and impractical for immediate field assessments. To address this limitation, the present study introduces an expedient approach for on-site analysis, employing fluorescence spectra obtained through excitation light irradiation of perilla leaves. Standard analytical techniques were employed to evaluate various constituent contents (chlorophyl (Chl), total polyphenol content (TPC), total flavonoid content (TFC), and rosmarinic acid (RA)) and functional attributes (DPPH radical scavenging activity, ferric reducing antioxidant power (FRAP), oxygen radical absorbance capacity (ORAC), and Aβ aggregation inhibitory activity). Correlations between the fluorescence spectra and these parameters were examined using normalized difference spectral index (NDSI), ratio spectral index (RSI), and difference spectral index (DSI) analyses. The resulting predictive model exhibited a high coefficient of determination, with *R*^2^ values equal to or greater than 0.57 for constituent contents and 0.49 for functional properties. This approach facilitates the convenient, simultaneous, and nondestructive monitoring of both the chemical constituents and the functional capabilities of perilla leaves, thereby simplifying the determination of optimal harvest times. The model derived from this method holds promise for real-time assessments, indicating its potential for the simultaneous evaluation of both constituents and functionalities in perilla leaves.

## 1. Introduction

*Perilla frutescens* has a long history of culinary and medicinal use in Asian countries, specifically as an antidote, antibiotic, and antipyretic [1]. Medicinal plants with antioxidant activity are still used as a source of medicines and nutraceuticals to reduce the action of free radicals and oxidative stress in the body [2,3,4]. The plant is rich in polyphenols and other bioactive compounds [5,6] and exhibits potent antioxidant [7], anti-inflammatory [8], and amyloid-β (Aβ) aggregation inhibitory activities [9]. These properties are particularly relevant to the pathogenesis of Alzheimer’s disease [10,11,12,13].

Among the polyphenols present in perilla and other Lamiaceae plants, 3-(3,4-dihydroxyphenyl)-2-{[(2E)-3-(3,4-dihydroxyphenyl)-2-propenoyl]oxy}propanoic acid (rosmarinic acid, RA) stands out for its multifaceted biological and pharmacological effects, including antioxidant, anti-inflammatory, antibacterial, antidepressant, and anticarcinogenic properties [14].

In earlier work, we conducted a comprehensive screening of over 500 natural resources and foods, primarily produced in Hokkaido, to identify those with Aβ aggregation inhibitory activity. Remarkably, *Perilla frutescens* var. *crispa*, sourced from Shiranuka, a town in Hokkaido, showed a remarkably high Aβ aggregation inhibitory activity [15]. The activity of perilla (EC_50_ = 0.48 µg/mL) was approximately 40 times greater than that of spearmint (EC_50_ = 18 µg/mL), the next most potent spice [16,17].

However, the biochemical composition and bioactivity of perilla leaves are subject to fluctuations influenced by both growth stage and environmental factors [18,19,20]. Traditional methods for quantifying these variables are labor-intensive and impractical for immediate, on-site assessments.

Previously reported on-site measurement techniques, such as imaging modalities employing hyperspectral satellite imagery, require sophisticated hardware and extensive data processing [21,22,23]. Additionally, existing handheld optical systems like Greenseeker™, atLEAF+, and SPAD502 are limited in their capabilities, only providing estimates of chlorophyll levels in small areas of 3 mm × 2 mm and at a high cost [24,25,26]. Prior studies that successfully correlated spectral data with component content typically utilized leaves from the same individual plant collected on the same day [27,28]. Therefore, further investigation is required to ascertain whether these spectral analysis methods can be adapted for broader applications, particularly for variables subject to environmental and seasonal changes, and whether they can be useful in determining optimal harvest times.

In this study, we aimed to establish a convenient, nondestructive, on-site measurement technique that would enable the simultaneous estimation of both the biochemical constituents and the functional attributes of *Perilla frutescens* leaves. The proposed method leverages fluorescence spectral data obtained by irradiating leaves with both ultraviolet and visible light [29]. This approach facilitates the simultaneous, nondestructive assessment of perilla leaves, offering a streamlined methodology for determining optimal harvest times.

## 2. Results

### 2.1. Fluorescence Spectra of Perilla Leaves

Figure 1 presents the 3D fluorescence spectra, recorded at excitation wavelengths ranging from 250 to 600 nm and fluorescence wavelengths from 260 to 800 nm. Observations indicate weak fluorescence in the 260–450 nm and 550–650 nm ranges when excited below 350 nm and strong fluorescence in the 650–800 nm range for excitation wavelengths above 350 nm.

### 2.2. Content of Constituents and Functionality of Perilla

Figure 2 shows the temporal variations in the concentrations of total chlorophyll (Chl), total polyphenol content (TPC), total flavonoid content (TFC), and RA, as well as functional properties such as DPPH radical scavenging activity, ferric reducing antioxidant power (FRAP), oxygen radical absorbance capacity (ORAC), and Aβ aggregation inhibitory activity. These variations were monitored during the weekly harvest of perilla leaves from 6 August to 19 November 2021. Notably, a decline in total Chl and TFC levels was observed from 1 October to 19 November. Detailed datasets corresponding to Figure 2 are provided in Appendix A.

### 2.3. Full-Wavelength R^2^ Contour Maps for NDSI, RSI, and DSI Analyses of Perilla

To further refine our analysis, contour maps were generated for each excitation light, covering a wavelength range of 250 to 600 nm at 5 nm intervals. Appendix A are animated graphics (GIFs) displaying the *R*^2^ values that correlate component indices with spectral indices as derived from the fluorescence spectra at varying excitation wavelengths. These analyses helped to identify the excitation wavelength that yielded the highest *R*^2^ value. Combinations of excitation and fluorescence wavelengths (*i*, *j*) that produced the highest coefficients of determination *R*^2^ and *R*^2^ are tabulated in Table 1 and illustrated in Figure 3, Figure 4 and Figure 5.

### 2.4. Correlation Modeling of Spectral Index and Component Index in Perilla

An estimation model was developed to predict various component indices—total Chl, TPC, TFC, RA, DPPH radical scavenging activity, FRAP, ORAC, and Aβ aggregation inhibitory activity—based on spectral indices (NDSI, RSI, and DSI) (Figure 6, Figure 7 and Figure 8). Linear regression analyses were conducted for each component index (Table 2). The outcomes revealed that the NDSI displayed the highest *R*^2^ values for RA and ORAC, the RSI was most effective for estimating Total Chl, TPC, TFC, and Aβ aggregation inhibitory activity, and the DSI was most suitable for DPPH radical scavenging activity and FRAP.

## 3. Discussion

The weak fluorescence observed in the 260–450 nm and 550–650 nm ranges with excitation below 350 nm (Figure 1) is attributed to the fluorescence of polyphenols covalently bound to the cell wall [30]. Conversely, the strong fluorescence observed in the 650–800 nm range at excitation wavelengths above 350 nm is ascribed to chlorophyll emission from the chloroplasts [29].

In plant physiology, it has been reported that chlorophyll levels tend to diminish during winter months [31], while flavonoid concentrations increase in response to light exposure [32]. Thus, the observed decline in flavonoid levels in this study (Figure 2) could be a consequence of reduced solar radiation during winter. Seasonal variations may also impact the concentration of polyphenols, particularly rosmarinic acid, in perilla leaves, which could in turn influence their functional properties from October onward [18,20]. Intriguingly, for indices such as Total Chl, RA, FRAP, ORAC, and Aβ aggregation inhibitory activity, the same optimal values were found for λ, ρi, and ρj across NDSI, RSI, and DSI. For other component indices, however, the maximum *R*^2^ was observed at different wavelengths depending on the spectral index (Figure 3, Figure 4 and Figure 5 and Table 1).

The findings presented in Figure 6, Figure 7 and Figure 8 suggest that the optimal spectral index (SI) varies depending on the specific component being measured. However, RSI emerged as the most effective method of analysis for perilla leaves across a majority of component indices. The lowest *R*^2^ value was 0.49 for ORAC when using NDSI. Given that ORAC primarily measures the effects of polyphenols, vitamin C, and vitamin E [33] and that the ORAC of perilla seed oil has been found to correlate more closely with total tocopherols than with polyphenols [34], this may account for the lower *R*^2^ value between SI and ORAC when based on the fluorescence of polyphenols and chlorophyll.

Prior studies that utilized fluorescence spectra for assessing the content of plant constituents have primarily focused on simplistic measures such as the increase or decrease in chlorophyll levels, identification of aberrant values, and variations in leaf water content [23,24,25,26]. Moreover, investigations into nitrogen (N) indices, including leaf N concentration (LNC), plant N concentration (PNC), plant N uptake (PNU), and N nutrition index (NNI), have reported a substantial correlation between fluorescence intensity ratios and these N indices, with *R*^2^ values ranging from 0.40 to 0.78 [35]. Techniques also exist for measuring polyphenols and other substances by absorbing carbon dots from root systems, but these approaches present labor and safety challenges [36]. Other studies have employed fluorescence imaging to monitor variations in leaf and fruit composition from the time of collection to shipping [29,37,38]. Nevertheless, to the best of our understanding, no established methodologies currently exist for concurrently predicting both the content of constituents and their functional attributes using fluorescence spectra, a gap this study aims to fill.

Studies deploying spectral methods have demonstrated limitations. For example, an assessment of the chlorophyll content in peanut leaves across an entire field, using hyperspectral data converted to a single-photon avalanche diode (SPAD) equivalent, yielded an *R*^2^ value of 0.69 when correlated with hyperspectral data [23]. This implies that the technique only provides a partial view, restricted to evaluating a single component and not accounting for variations in chlorophyll content across individual leaves. Similarly, an NDSI analysis of carnosic acid in rosemary using near-infrared spectra reported a correlation coefficient (*r*) of 0.81 [27], which translates to an *R*^2^ value of 0.6561—lower than the coefficient of determination for RA in our study. Additionally, although the sample size in that study was larger (*n* = 79), the samples were collected on a single day, reducing the method’s applicability for determining optimal harvesting times. In an analysis of antioxidant activity, correlation coefficients of *r* = 0.86 and *r* = 0.85 were obtained for DPPH radical scavenging activity and FRAP, respectively, using the near-infrared spectra of bamboo leaf extracts [39]. Although these correlations were higher than those found in our study, our approach offers the distinct advantage of enabling a nondestructive evaluation of both constituent content and multiple functionalities, including antioxidant activity and Aβ aggregation inhibitory activity. This constitutes a more versatile tool for comprehensive plant analysis.

While the predictive capabilities of the spectral model derived from this study may not be the most robust, the method nonetheless offers the distinct advantage of enabling nondestructive assessments of various plant functionalities, including antioxidant and Aβ aggregation inhibitory activities. Moreover, the correlation coefficients for total polyphenol content in relation to DPPH radical scavenging activity, FRAP, and Aβ aggregation inhibitory activity were recorded as 0.60, 0.59, and 0.47, respectively (Appendix A). These findings underscore the existence of correlations not only between the concentration of total polyphenols and antioxidant activities but also between the fluorescence spectra elicited by excitation light irradiation and these biochemical properties. Importantly, this analytical method is less susceptible to fluctuations in the environmental conditions or measurement apparatus due to its reliance on ratiometry [40]. However, several caveats must be noted that could limit the general applicability of a spectral model based on a specific dataset for predicting new properties.

One such limitation is the potential need for customized models tailored to specific applications or plant types. For instance, perilla plants exist in various shades of red and green, with these color variations attributed to anthocyanin content [41]. Given that anthocyanins are known to exhibit fluorescence around 600 nm [42] and that the emission profiles can differ depending on the specific polyphenols present [29], researchers intending to implement this model in practical settings may find it necessary to develop their own correction factors to account for these variations.

The second complicating factor is the variability introduced by the apparatus used for fluorescence measurement. Various pre-optical devices such as integrating spheres, contact probes, and leaf clips can be employed to measure leaves, each introducing potential variations in observed geometry and anisotropic surface reflectance [43]. As a result, the utility of the model generated from this study could be compromised unless measurements are made on a standardized 1 cm² leaf area using a consistent leaf clip or similar device.

The third challenge arises from the nuances in commercial fluorescence spectrometers and their associated software, which may employ slightly divergent sensor technologies and spectral processing algorithms [44]. Consequently, researchers might need to undertake calibration procedures to harmonize fluorescence spectra acquired from different instruments.

Despite these considerations, the linear models developed in this study demonstrated satisfactory predictive accuracy. A predictive model for component content with a high coefficient of determination, *R*^2^ ≥ 0.57, and another for functional properties with *R*^2^ ≥ 0.49 were identified. These high coefficients of determination indicate that robust single-variable estimation models for each index were successfully constructed.

## 4. Materials and Methods

### 4.1. Chemicals and Equipment

Chemicals, including methanol (HPLC grade), acetone (reagent grade), ethanol (99.5%, reagent grade), acetic acid (reagent grade), potassium hexacyanoferrate (III), potassium hydrogen phosphate, sodium hydrogen phosphate, dimethyl sulfoxide were purchased from Kanto Chemical Co. Inc. (Tokyo, Japan). 2,2′-Azobis-2-methyl-propanimidamide, dihydrochloride (AAPH), 1,1-diphenyl-2-picrylhydrazine free radical (DPPH), 2-(3,4-dihydroxyphenyl)-3,5,7-trihydroxychromen-4-one (quercetin), and 3,4,5-trihydroxybenzoic acid (gallic acid) were purchased from Tokyo Chemical Industry Co. Ltd. (Tokyo, Japan). Formic acid, aluminum (III) chloride hexahydrate, sodium dodecyl sulfate, and L(+)-ascorbic acid were purchased from Fujifilm Wako Pure Chemical Co. Ltd. (Osaka, Japan). RA, Trolox, and Folin–Ciocalteu phenol Reagent (2N) were purchased from Sigma-Aldrich (St. Louis, MO, USA). Sodium bicarbonate was purchased from Junsei Chemical Co., Ltd. (Tokyo, Japan).

For spectral analysis, a spectrofluorophotometer FP-8500 from JASCO Corp. (Tokyo, Japan) was utilized to record photoluminescence spectra. UV–visible absorbance and fluorescence measurements were conducted in 96-well plates using the SH-9000Lab apparatus from Corona Electric Co., Ltd. (Hitachinaka, Japan). Liquid chromatography–mass spectrometry (LC-MS) analyses were executed with a Shimadzu LCMS-8045, equipped with an LC-2060C UV detector, electrospray ionization (ESI), and a triple quadrupole (Shimadzu, Kyoto, Japan).

### 4.2. Perilla Samples

This study involved the growth of *Perilla frutescens* var. *crispa* at our university field, with seedlings originally cultivated at AW Farm Chitose (Chitose, Japan). This study focused on leaves harvested from the fourth node, with three leaves collected weekly (*n* = 40). These leaves were subsequently subjected to fluorescence spectra and chlorophyll measurements. Concurrently, the remaining leaf tissue was extracted using 99.5% ethanol at a volume five times that of the leaf material. The extraction was performed for three weeks, and the samples were stored at 4 °C until further analysis. Based on prior assessments, it was concluded that perillaldehyde, the primary component of perilla, would reach saturation after a one-week extraction period. Therefore, a three-week extraction window was deemed acceptable. This observation is documented as Appendix A.

### 4.3. Fluorescence Spectrum Measurement

In this study, the posterior end of the leaf was selected as the site for fluorescence measurements, as illustrated in Figure 9. This choice was based on reports indicating a higher concentration of polyphenols close to the stem in tobacco leaves [29]. To verify this, fluorescence at 690 nm and 740 nm—the characteristic wavelengths for chlorophyll fluorescence—was measured by placing a perilla leaf on a 96-well plate (Appendix A). The recorded values (416.0 ± 77.0, 1391.0 ± 206.8) were statistically comparable to the average fluorescence intensity for the whole leaf (394.7 ± 113.9, 1287.3 ± 360.0), as detailed in Appendix A. For these measurements, the excitation light was incident at an angle of 30° on the leaf surface. The wavelength of the excitation light varied between 260 nm and 800 nm in 5 nm increments, and fluorescence spectra were recorded every 0.5 nm [29]. These data were subsequently adjusted to account for variations in the sensitivity of the spectrofluorophotometer’s light detector across different wavelengths.

### 4.4. Chlorophyll (Chl) Quantification

The Chl content was quantified using the Porra method [45]. The leaves utilized for fluorescence spectral analysis were homogenized in acetone. The resulting suspension was centrifuged for 5 min, and the supernatant was transferred to another PCR tube. This procedure was repeated until the green pigment was entirely removed from the precipitate. To adjust the acetone concentration to 80%, ultrapure water was added to the supernatant. The spectrophotometric absorbance of the resultant solution was measured at wavelengths of 663.6 nm, 646.6 nm, and 750 nm. The wavelengths 663.6 nm and 646.6 nm correspond to the maximum absorption peaks for Chl a and Chl b in an 80% acetone solution, respectively. The absorbance at 750 nm served as the background measurement and was subtracted from the other absorbance values. The Chl concentration was then calculated using Equations (1)–(3).
Chl a (µg/mL) = 12.25 × A_663.6_ − 2.85 × A_646.6_.(1)
Chl b (µg/mL) = 20.31 × A_646.6_ − 4.91 × A_663.6_.(2)
Total Chl = Chl a + Chl b.(3)

### 4.5. Total Polyphenol Content (TPC)

The total polyphenol content in each extract was quantified utilizing the Folin–Ciocalteu method [46]. Perilla extracts obtained through extraction with 99.5% ethanol were subjected to reduced-pressure evaporation and then dissolved in 70% methanol at concentrations of 1000, 500, and 250 µg/mL. Each prepared extract (80 µL) was combined with Folin–Ciocalteu reagent (0.2 N, 400 µL) and allowed to stand at ambient temperature for 5 min. Subsequently, a sodium bicarbonate solution (7.5%, 320 µL) was added. After a 2 h incubation at 30 °C, the absorbance at 760 nm was measured. A calibration curve was constructed using gallic acid as the standard, allowing the polyphenol content to be calculated in terms of gallic acid equivalents (GAEs).

### 4.6. Total Flavonoid Content (TFC)

For the assessment of total flavonoids, the AlCl_3_ colorimetric method was employed [47]. Samples of perilla extract were evaporated under reduced pressure and dissolved in methanol at concentrations of 1000, 500, and 250 µg/mL. Each extract (400 µL) was mixed with a 2% AlCl_3_ solution (400 µL) and incubated for 10 min at 30 °C. The absorbance at 415 nm was then measured. A calibration curve, constructed using quercetin as the standard, facilitated the calculation of flavonoid concentrations in terms of quercetin equivalents (QEs).

### 4.7. RA Quantification by LC-MS

LC-MS analysis was performed using an LCMS-8045 coupled with an LC-2060C UV detector and an ESI source, both from Shimadzu, Kyoto, Japan. The analytical column used was a Kinetex^®^ C18 (2.1 × 150 mm, 5 µm, Phenomenex, Torrance, CA, USA) maintained at 40 °C. Detection occurred at a wavelength of 254 nm. The mobile phases employed were 0.1% formic acid in a 10% methanol/water mixture (A) and 0.1% formic acid in methanol (B). The gradient program initiated with 100% A and was adjusted according to a predefined schedule. The injection volume was set at 5 µL. The gradient with time was as follows (A to B): 0 min, 100:0; 4 min, 80:20; 9 min, 75:25; 14 min, 62:38; 17 min, 50:50; 21 min, 30:70; 22 min, 25:75; 35 min, 0:100; 45 min, 100:0. The mass spectrometric conditions were specified as follows: nebulizer gas flow at 3 L/min, heating gas flow at 10 L/min, interface temperature at 300 °C, DL temperature at 250 °C, and heat block temperature at 400 °C. The draining gas flow was maintained at 10 L/min. Precursor and product ions were monitored at *m*/*z* 359, 161, and 197, respectively, under selected reaction monitoring (SRM) mode.

The calibration curves for RA were constructed within the concentration range of 0.1 to 1.0 μg/mL, as depicted in Appendix A. The standards were processed identically to the perilla samples, achieving linearity with a correlation coefficient of 0.92. All the analyzed samples were encompassed within the 95% confidence interval of the calibration curve.

### 4.8. Assessment of DPPH Radical Scavenging Activity [48]

Perilla extracts, obtained by extraction with 99.5% ethanol, were subjected to reduced-pressure evaporation and then redissolved in ethanol at concentrations ranging from 7.8125 to 1000 µg/mL. Each sample (400 µL) was combined with 0.01 mM DPPH dissolved in 99.5% ethanol (400 µL). Next, 200 µL from each mixture was transferred to a 96-well microplate reader and incubated at 30 °C for 2 h. The absorbance was measured at a wavelength of 517 nm. Subsequently, the inhibition rate of DPPH oxidation was calculated using Equation (4). The IC_50_ values were deduced from the inhibition curves by plotting the inhibition rate against the sample concentration.
(4)$$Inhibition\;rate\;\left(\%\right)=\frac{A_{control}-A_{sample}}{A_{control}}\times 100.$$

### 4.9. Assessment of Ferric Reducing Antioxidant Power (FRAP)

FRAP was determined via a modified Oyaizu method [49,50]. Perilla extracts were evaporated under reduced pressure and redissolved in ethanol to attain a concentration of 1 mg/mL, which was further diluted to 500, 250, and 125 µg/mL. To 8 µL of each dilution, a mixture of 99.5% ethanol (72 µL), ultrapure water (400 µL), 1 M HCl (120 µL), 1% K_3_[Fe(CN)_6_] (120 µL), 1% sodium dodecyl sulfate (SDS) (40 µL), and 2% FeCl_2_ (40 µL) was added. This composite was incubated in a water bath at 50 °C for 20 min, then cooled to room temperature, and thoroughly mixed. The reducing capacity of each extract was assessed by measuring the increase in absorbance at 750 nm. Ascorbic acid served as the positive control, and the FRAP was calculated in terms of ascorbic acid equivalents (AAEs).

### 4.10. Oxygen Radical Absorbance Capacity (ORAC) [51]

An amount of 500 µL of the ethanol extract was subjected to reduced-pressure evaporation, and the resultant residue was redissolved in an AWA solution—comprising acetone, ultrapure water, and acetic acid in the ratio 70:29.5:0.5—to achieve a concentration of 12.5 µg/mL. These samples were further diluted with an assay buffer—consisting of 75 mM K_2_HPO_4_ and 75 mM KH_2_PO_4_, adjusted to a pH of 7.4—to concentrations of 12.5, 6.25, 3.125, and 1.5625 µg/mL. To 20 µL of each sample or assay buffer (as control), 200 µL of fluorescein (FL) working solution—constituted at 36 ng/mL of FL in assay buffer—was added. This mixture was subjected to shaking and stirring at 37 °C in a 96-well microplate reader, and its fluorescence intensity was measured at an emission wavelength (E_m_) of 520 nm and an excitation wavelength (E_x_) of 485 nm. These initial readings were designated as f_0min_. Subsequently, after a 10 min incubation at 37 °C, 75 µL of 2,2′-azobis(2-methylpropionamidine) dihydrochloride (AAPH) solution—prepared at 8.6 mg/mL of AAPH in assay buffer—was added. The fluorescence intensity at Em: 520 nm was recorded every 2 min for a duration of 90 min (f_2min_ to f_90min_). The time immediately preceding the addition of AAPH was defined as f_0min_.

Standard solutions of Trolox were prepared at concentrations of 12.5, 6.25, 3.125, and 1.5625 µg/mL.
AUC = 2 (0.5 × f_8min_ + f_10min_ + f_12min_ + f_14min_ +……+ f_88min_ + 0.5 × f_90min_)/f_0min_.(5)
netAUC_Trolox_ = AUC_Trolox_ + AUC_Blank_.(6)
netAUC_sample_ = AUC_sample_ + AUC_Blank_.(7)

A quadratic regression equation (y = ax^2^ + bx + c) was derived using Trolox concentrations on the *x*-axis and the net area under the curve for Trolox (AUC_Trolox_) on the *y*-axis.

The ORAC value was then calculated using Equation (8).
(8)$$ORAC\left(\frac{µmol\;TE}{g}\right)=\frac{\left(a\times {\left({netAUC}_{sample}\right)}^{2}+b\times \left({netAUC}_{sample}\right)+c\right)\times V\times D}{W},$$
where TE refers to Trolox equivalent, while a, b, c are the coefficients in the quadratic regression equation. Additional variables include V, the volume of the sample stock solution; D, the dilution factor of the sample stock solution; and W, the weight of the sample.

### 4.11. Evaluation of Amyloid-Beta Aggregation Inhibitory Activity Using the Automated Microliter-Scale High-Throughput Screening (MSHTS) System [16,52]

For the microliter-scale high-throughput screening (MSHTS) assay, human Aβ_42_ was acquired from Peptide Institute Inc. (Osaka, Japan). The quantum-dot (QD) Aβ nanoprobe was synthesized using QD-PEG-NH_2_ (from Thermo Fisher Scientific, Waltham, MA, USA) and Cys-conjugated Aβ_40_ (obtained from Anaspec Inc., Fremont, CA, USA). The half-maximal effective concentration (EC_50_) for each plant extract was evaluated using an adapted automated MSHTS system. Specifically, solutions were prepared with plant extracts at six concentrations, combined with 25 nM QDAβ and 25 μM Aβ_42_ in a phosphate-buffered saline (PBS) solution containing 5% ethanol and 2.5% dimethyl sulfoxide. These mixtures were dispensed into a 1536-well plate (782096, Greiner, Kremsmünster, Austria) and incubated at 37 °C for 24 h. Fluorescence images of each well were captured pre- and postincubation using an inverted fluorescence microscope (ECLIPSE Ti-E; Nikon, Tokyo, Japan) equipped with a color CMOS camera (DS-Ri2; Nikon). QD fluorescence was photographed using a 4× objective lens (Plan Apo λ 4×/0.2, Nikon) and a tetramethylrhodamine isothiocyanate (TRITC) filter set (TRITC-A-Basic-NTE, Semrock, Rochester, NY, USA). The standard deviation (SD) of the fluorescence intensity within the central region of interest in each well was computed using the General Analysis program NIS-Elements. Samples with the highest concentration of plant extract, which could potentially contain insoluble substances, were excluded, as these interfered with SD calculations and thus compromised the accuracy of the EC_50_ estimation. The EC_50_ values were subsequently determined from these SD values through the Prism software version 8.4.3 (GraphPad Software, San Diego, CA, USA), employing a global fit to an asymmetric sigmoidal, five-parameter logistic model.

### 4.12. Spectrum Analysis

Figure 10 presents a schematic representation of the spectrum analysis approach. Various spectral indices, such as the normalized difference spectral index (NDSI), ratio spectral index (RSI), and difference spectral index (DSI), are computed from the 3D map of fluorescence spectra corresponding to samples of perilla leaves at each excitation wavelength. These indices are designed to identify the optimal spectral characteristics associated with the chemical composition and functional attributes of the samples [21,22]. The NDSI, RSI, and DSI are defined as follows:(9)$$NDSI_{\lambda }\left[i,j\right]=\frac{\left(\rho_{i}-\rho_{j}\right)}{\left(\rho_{i}+\rho_{j}\right)},$$
(10)$$\;\mathrm{RS}I_{\lambda }\left[i,j\right]=\frac{\rho_{i}}{\rho_{j}},$$
(11)$$\mathrm{DS}I_{\lambda }\left[i,j\right]=\rho_{i}-\rho_{j},$$
where *i* and *j* refer to specific fluorescence wavelengths, ranging from 260 to 800 nm. The term *ρ_i_* represents the fluorescence intensity at the wavelength *i*, with the excitation wavelength λ varying between 250 and 600 nm. Calculations for the NDSI, RSI, and DSI were performed for all combinations of fluorescence intensities at the examined excitation wavelengths.

The relationship between the spectral indices (NDSI, RSI, and DSI values), denoted as SI, and the component indices (which quantitatively represent the content of constituents and functionality), denoted as CI, was investigated using the square of the Pearson product-moment correlation coefficient, *R*^2^. This coefficient is expressed mathematically as follows:(12)$$R^{2}={\left(\frac{\sum_{k=1}^{N}\left(SI_{k}-\bar{SI}\right)\left(CI_{k}-\bar{CI}\right)}{\sqrt{\sum_{k=1}^{N}{\left(SI_{k}-\bar{SI}\right)}^{2}}\sqrt{\sum_{k=1}^{N}{\left(CI_{k}-\bar{CI}\right)}^{2}}}\right)}^{2},$$
where SI*_k_* and CI*_k_* denote the value of SI and CI obtained from the *k*th leaf sample, and the *N* denotes the total number of the leaf samples. The corresponding averages are denoted by $\bar{SI}$ and $\bar{CI}$. The SI was calculated from the combination of any two wavelengths of the fluorescence spectrum ranging between 260 and 800 nm, and its correlation with CI was investigated.

To quantify this relationship, the Pearson product-moment correlation coefficient was utilized, comparing the numerical data obtained from the spectral index calculations for all fluorescence wavelength combinations with the component index data for each leaf sample. The resulting *R*^2^ values signify the strength of the correlation between the spectral index and the component index. These values are graphically represented in a contour map, where the vertical axis corresponds to the fluorescence intensity at wavelength i (*ρ_i_*), and the horizontal axis pertains to the fluorescence intensity at wavelength *j* (*ρ_j_*). Figure 10 employs a color scheme to indicate the magnitude of *R*^2^; the hue transitions from blue to red as the correlation coefficient increases, signifying a stronger correlation between the spectral index and the component index for those specific combinations of fluorescence intensities.

## 5. Conclusions

In this study, regression models with a coefficient of determination (*R*^2^) of 0.49 or higher were successfully employed to identify two fluorescence-spectra-based indices. These indices are capable of predicting both the content of constituents and the functionalities of perilla leaves. This accomplishment enables the convenient, simultaneous, and nondestructive assessment of perilla leaves, facilitating straightforward harvest-time diagnostics.

While the model generated through this methodology is both uncomplicated and practical, it should be noted that this research was conducted under laboratory conditions. Consequently, future work must focus on designing a portable and cost-effective sensor to enable on-site implementation of this technological approach. Additionally, as this study was confined to a specific geographical locale, the extension of this research to other regions or varied geographical and climatic conditions is requisite for its practical application.

## Figures and Tables

**Figure 1 molecules-28-07199-f001:**
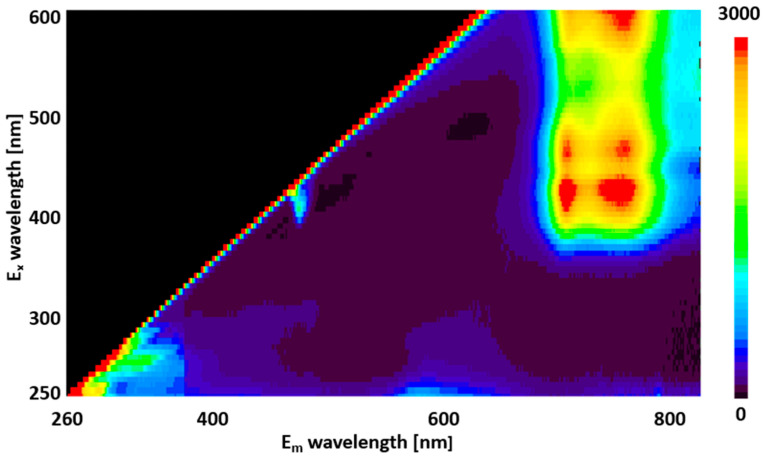
Three-dimensional representation of fluorescence spectra, ranging from 260 to 800 nm, generated upon irradiating perilla leaves with excitation light between 250 and 600 nm. The vertical axis depicts the excitation light wavelength, while the horizontal axis displays the fluorescence wavelength. Color gradation represents fluorescence intensity, transitioning from black (absence of fluorescence emission) to red (3000 a.u.).

**Figure 2 molecules-28-07199-f002:**
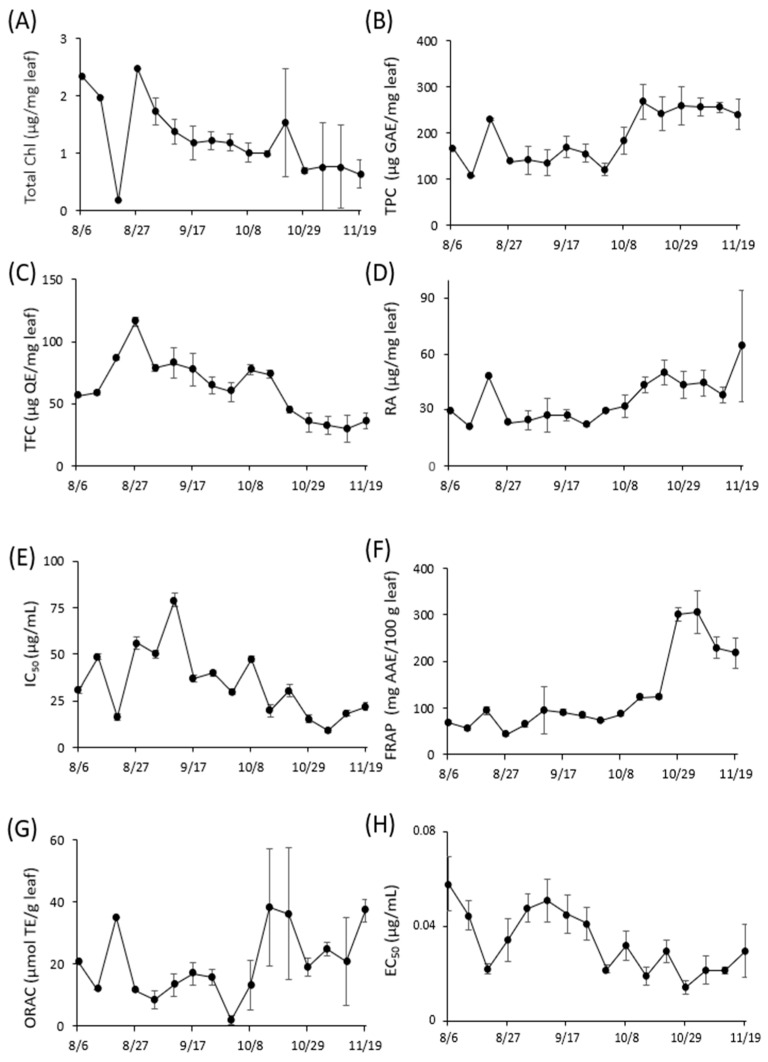
Variation in constituent content and functional properties of *Perilla frutescens* leaves. (**A**): Total Chl, (**B**): TPC, (**C**): TFC, (**D**): RA, (**E**): DPPH radical scavenging activity, (**F**): FRAP, (**G**): ORAC, (**H**): Aβ aggregation inhibitory activity. Each data point represents the mean of triplicate measurements, and the error bars indicate standard error of mean (SEM).

**Figure 3 molecules-28-07199-f003:**
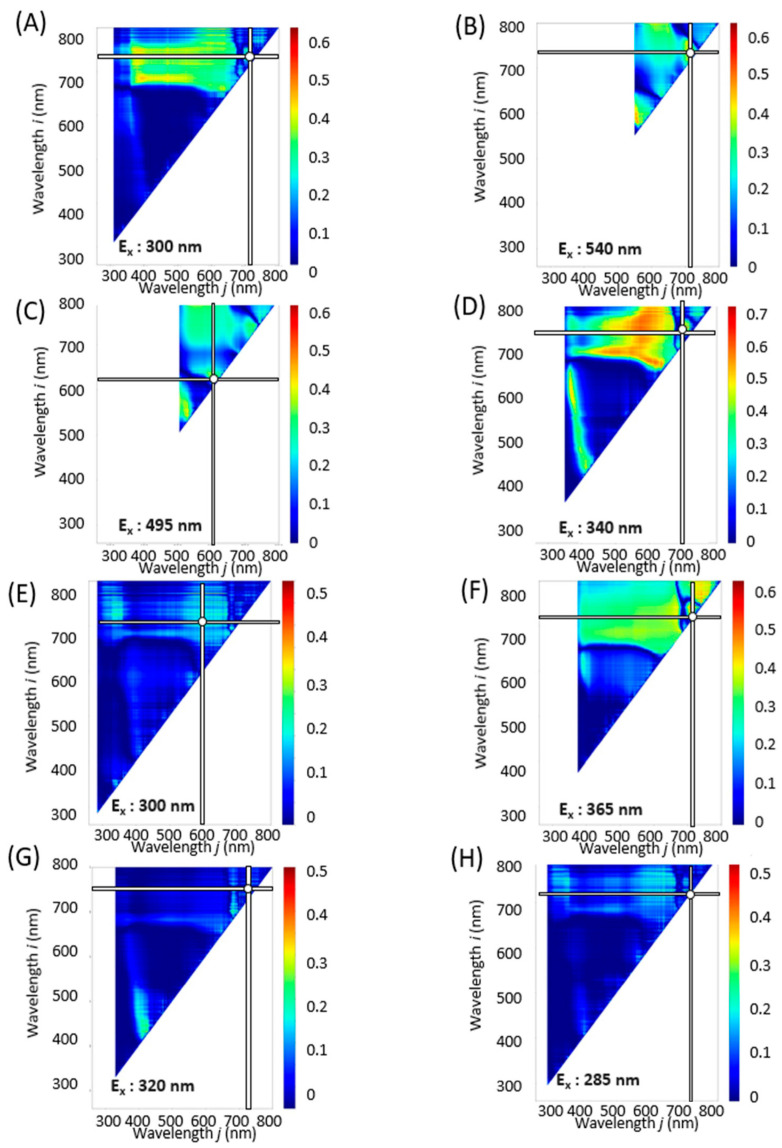
NDSI contour maps corresponding to constituent contents and functionalities based on combinations of excitation and fluorescence wavelengths (*i*, *j*) with the highest *R*^2^ (E_x_, E_m*i*_, E_m*j*_) value and coefficient of determination (*R*^2^). (**A**): Total Chl, (**B**): TPC, (**C**): TFC, (**D**): RA. (**E**): DPPH radical scavenging activity, (**F**): FRAP, (**G**): ORAC, (**H**): Aβ aggregation inhibitory activity. The horizontal and vertical bars on the spectrum indicate the fluorescence wavelengths with the highest *R*^2^ values.

**Figure 4 molecules-28-07199-f004:**
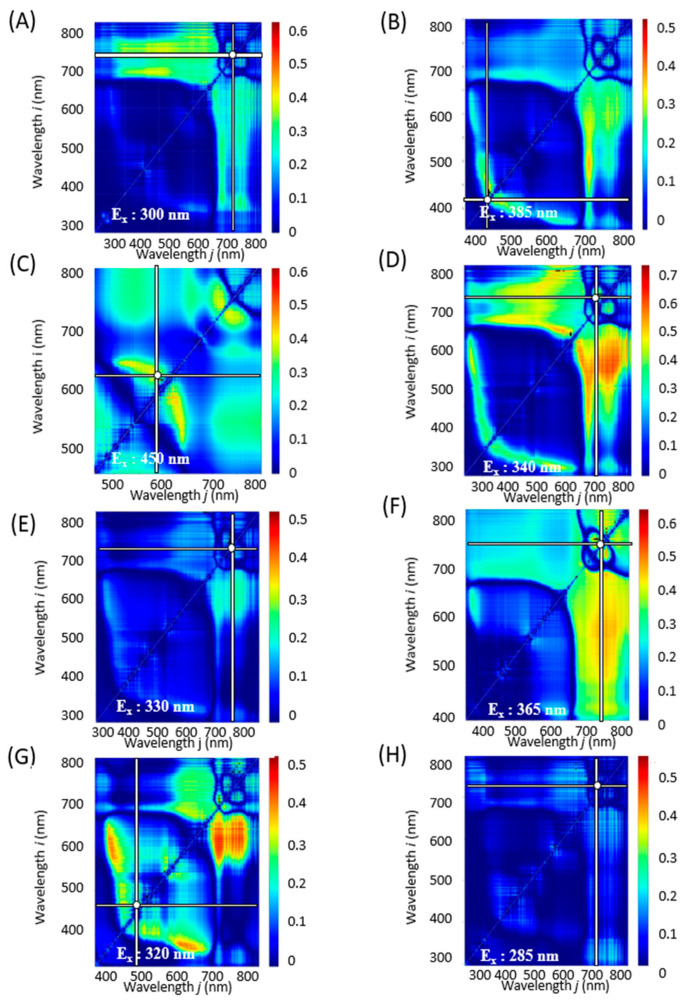
RSI contour maps corresponding to constituent contents and functionalities with the combinations of excitation and fluorescence wavelengths (*i*, *j*) with the highest *R*^2^ (E_x_, E_m*i*_, E_m*j*_) and coefficient of determination (*R*^2^). (**A**): Total Chl, (**B**): TPC, (**C**): TFC, (**D**): RA. (**E**): DPPH radical scavenging activity, (**F**): FRAP, (**G**): ORAC, (**H**): Aβ aggregation inhibitory activity. The horizontal and vertical bars on the spectrum indicate the fluorescence wavelengths with the highest *R*^2^ values.

**Figure 5 molecules-28-07199-f005:**
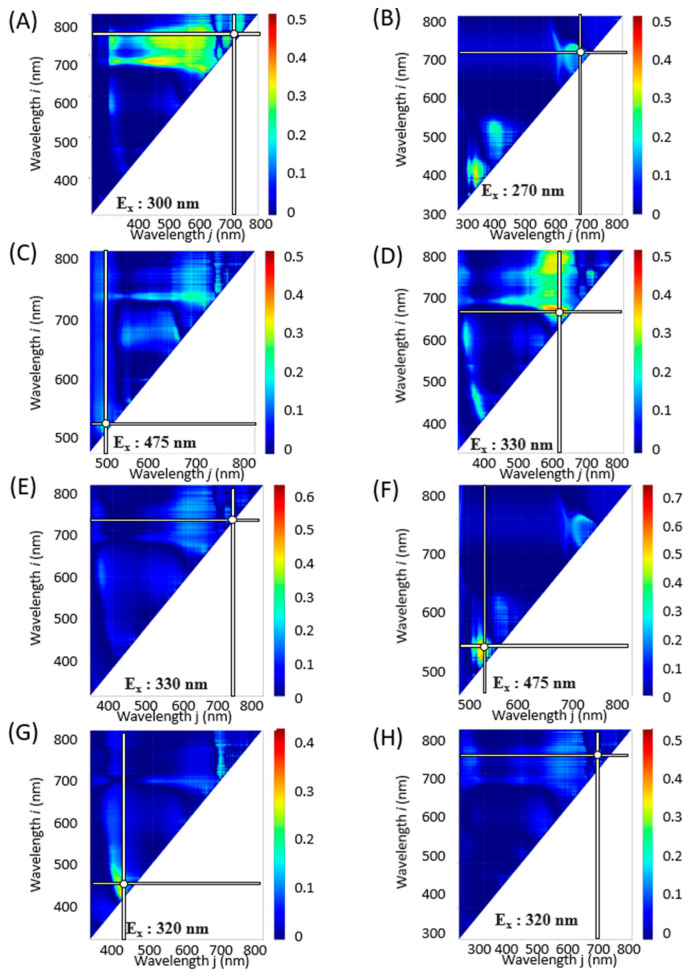
DSI contour maps corresponding to constituent contents and functionalities with the combinations of excitation and fluorescence wavelengths (*i*, *j*) with the highest *R*^2^ (E_x_, E_m*i*_, E_m*j*_) and coefficient of determination (*R*^2^). (**A**): Total Chl, (**B**): TPC, (**C**): TFC, (**D**): RA. (**E**): DPPH radical scavenging activity, (**F**): FRAP, (**G**): ORAC, (**H**): Aβ aggregation inhibitory activity. The horizontal and vertical bars on the spectrum indicate the fluorescence wavelengths with the highest *R*^2^ values.

**Figure 6 molecules-28-07199-f006:**
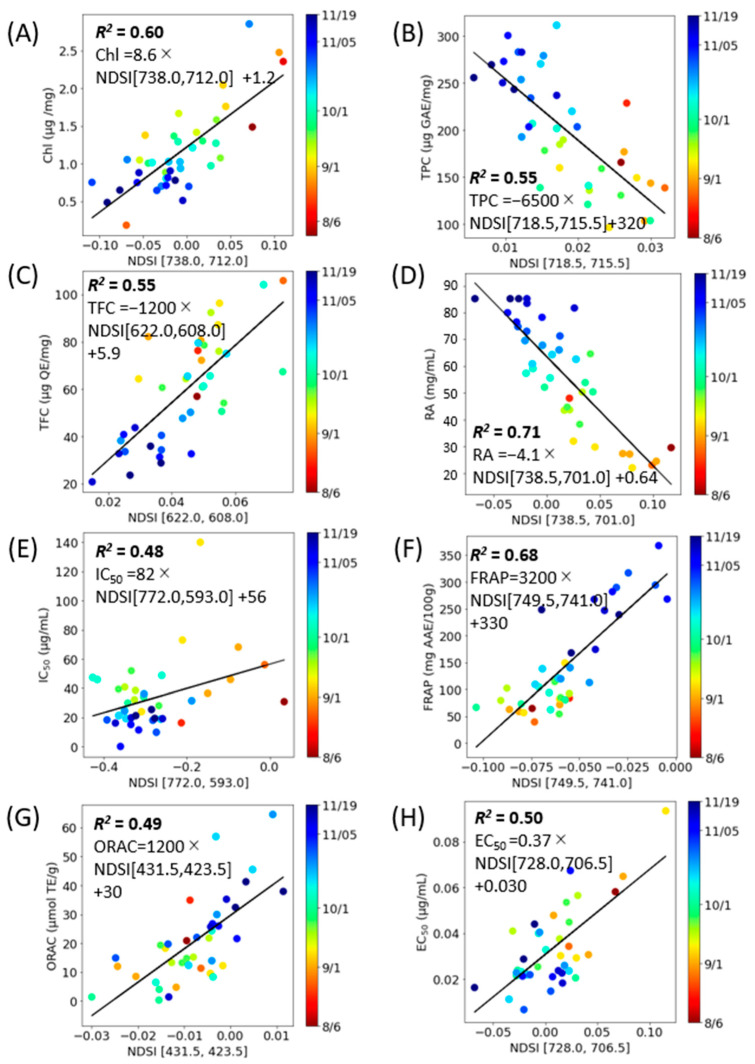
NDSI correlation model corresponding to constituent contents with the combinations of excitation and fluorescence wavelengths (*i*, *j*) with the highest *R*^2^ (E_x_, E_m*i*_, E_m*j*_) and coefficient of determination (*R*^2^). (**A**): Total Chl, (**B**): TPC, (**C**): TFC, (**D**): RA, (**E**): DPPH radical scavenging activity, (**F**): FRAP, (**G**): ORAC, (**H**): Aβ aggregation inhibitory activity. Each data point represents the mean of triplicate measurements. The solid line depicts the regression line.

**Figure 7 molecules-28-07199-f007:**
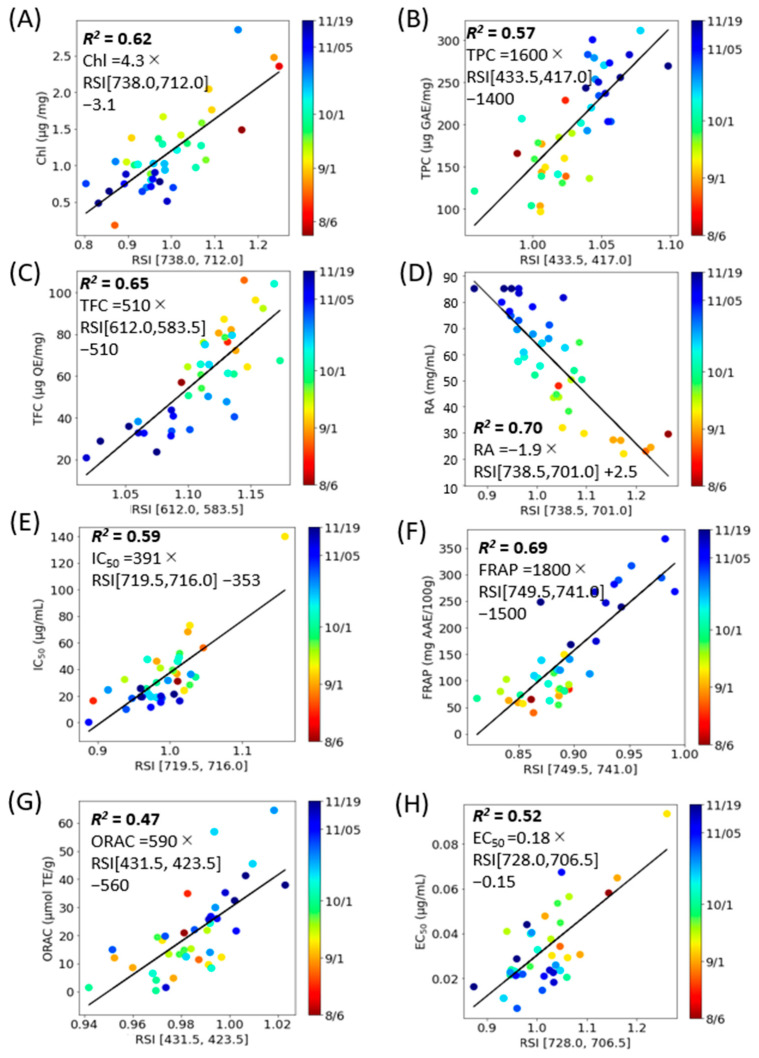
RSI correlation model corresponding to constituent contents with combinations of excitation and fluorescence wavelengths (*i*, *j*) with the highest *R*^2^ (E_x_, E_m*i*_, E_m*j*_) and coefficient of determination (*R*^2^). (**A**): Total Chl, (**B**): TPC, (**C**): TFC, (**D**): RA, (**E**): DPPH radical scavenging activity, (**F**): FRAP, (**G**): ORAC, (**H**): Aβ aggregation inhibitory activity. Each data point represents the mean of triplicate measurements. The solid line depicts the regression line.

**Figure 8 molecules-28-07199-f008:**
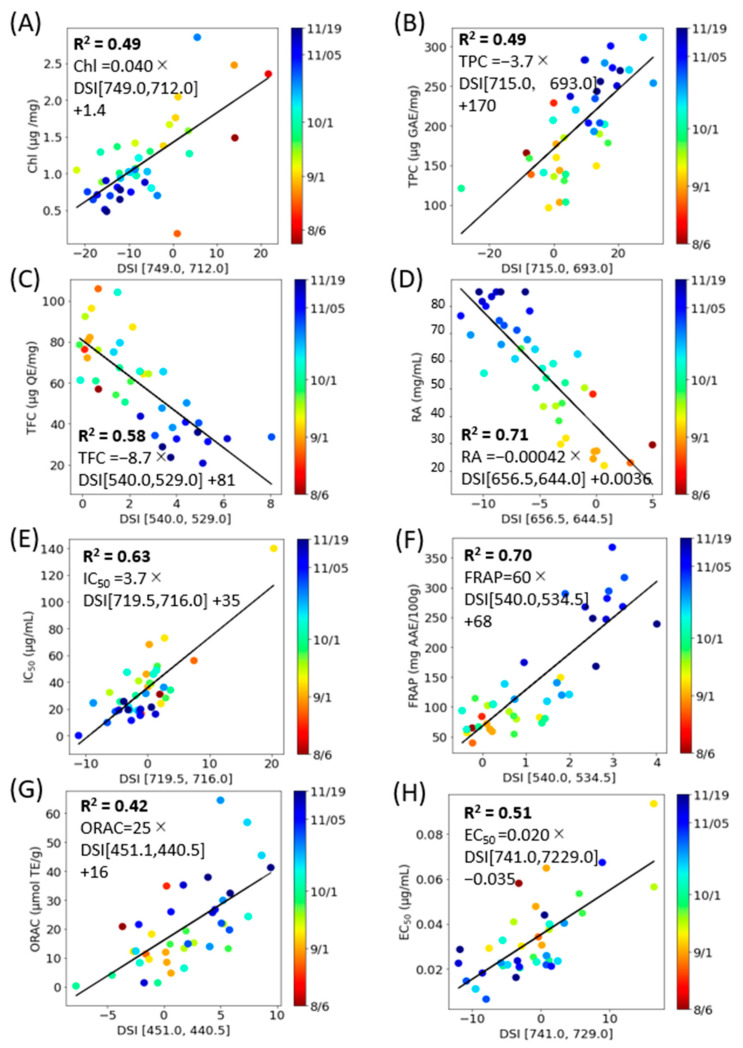
DSI correlation model corresponding to constituent contents with the combinations of excitation and fluorescence wavelengths (*i*, *j*) with the highest *R*^2^ (E_x_, E_m*i*_, E_m*j*_) and coefficient of determination (*R*^2^). (**A**): Total Chl, (**B**): TPC, (**C**): TFC, (**D**): RA, (**E**): DPPH radical scavenging activity, (**F**): FRAP, (**G**): ORAC, (**H**): Aβ aggregation inhibitory activity. Each data point represents the mean of triplicate measurements. The solid line depicts the regression line.

**Figure 9 molecules-28-07199-f009:**
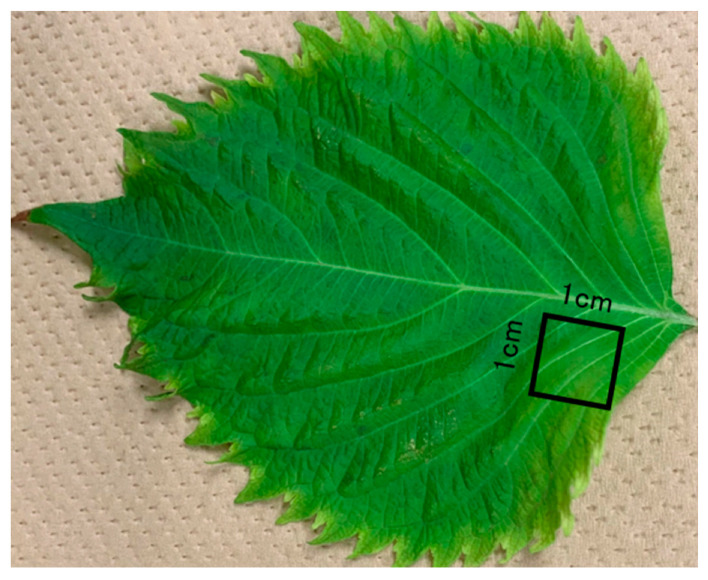
Leaf area from which fluorescence spectra were obtained in this study.

**Figure 10 molecules-28-07199-f010:**
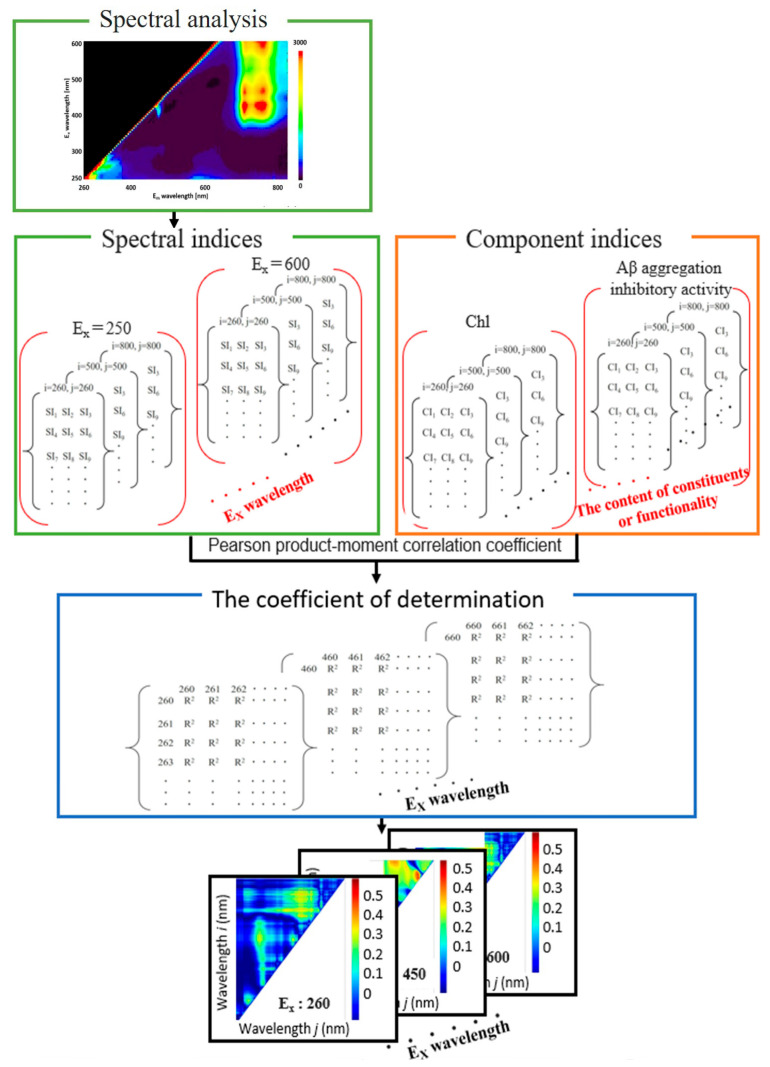
Methodological steps for calculating the coefficients of determination for both spectral and component indices and for generating the associated maps.

**Table 1 molecules-28-07199-t001:** Identification of excitation and fluorescence wavelengths (*i*, *j*) associated with the highest coefficient of determination *R*^2^ for each component index.

Index	NDSI				RSI				DSI			
	λ	*i*	*j*	*R* ^2^	λ	*i*	*j*	*R* ^2^	λ	*i*	*j*	*R* ^2^
	(nm)	(nm)	(nm)		(nm)	(nm)	(nm)		(nm)	(nm)	(nm)	
Constituents												
Total Chl	300	738	712	0.60	300	738	712	0.62	300	749	712	0.49
TPC	540	718.5	715.5	0.55	385	433.5	417	0.57	270	715	693	0.49
TFC	495	622	608	0.55	450	612	583.5	0.65	475	540	529	0.58
RA	340	738.5	701	0.71	340	738.5	701	0.70	330	656.5	644.5	0.71
Functionality												
DPPH	300	772	593	0.48	330	719.5	716	0.59	330	719.5	716	0.63
FRAP	365	749.5	741	0.68	365	749.5	741	0.69	475	540	534.5	0.70
ORAC	320	772	593	0.48	320	486.5	476	0.47	255	451	440.5	0.42
Aβ	285	728	706.5	0.50	285	728	706.5	0.52	320	741	729	0.51

**Table 2 molecules-28-07199-t002:** Regression line equation and corresponding coefficient of determination for the correlation models.

	NDSI		RSI		DSI	
	Regression Line	*R* ^2^	Regression Line	*R* ^2^	Regression Line	*R* ^2^
Constituents						
Total Chl (µg/mg)	Chl = 8.6 × NDSI_300_[738.0, 712.0] + 1.2	0.60	Chl = 4.3 × RSI_300_[738.0, 712.0] − 3.1	0.62	Chl = 0.040 × DSI_300_[749.0, 712.0] + 1.4	0.49
TPC (µg GAE/mg)	TPC = –6500 × NDSI_540_[718.5, 715.5] + 320	0.55	TPC = 1600 × RSI_385_[433.5, 417.0] − 1400	0.57	TPC = 3.7 × DSI_270_[715.0, 693.0] + 170	0.49
TFC (µg QE/mg)	TFC = –1200 × NDSI_495_[622.0, 608.0] + 5.9	0.55	TFC = 510 × RSI_450_[612.0, 583.5] − 510	0.65	TFC = –8.7 × DSI_475_[540.0, 529.0] + 81	0.58
RA (µg/mg)	RA = –4.1 × NDSI_340_[738.5, 701.0] + 0.64	0.71	RA = –1.9 × RSI_340_[738.0, 701.0] + 2.5	0.70	RA = –0.042 × DSI_330_[656.5, 644.0] + 0.36	0.71
Functionality						
DPPH (µg/mL)	IC_50_ = 82 × NDSI_300_[772.0, 593.0] + 56	0.48	IC_50_ = 391 × RSI_330_[719.5, 716.0] − 353	0.59	IC_50_ = 3.7 × DSI_330_[719.5, 716.0] + 35	0.63
FRAP (mg AEE/100 g)	FRAP = 3200 × NDSI_365_[749.5, 741.0] + 330	0.68	FRAP = 1800 × RSI_365_[749.5, 741.0] − 1500	0.69	FRAP = 60 × DSI_475_[540.0, 534.5] + 68	0.70
ORAC (µmol TE/g)	ORAC = 1200 × NDSI_320_[772.0, 593.0] + 30	0.49	ORAC = 590 × RSI_320_[486.5, 476.0] − 560	0.47	ORAC = 25 × DSI_320_[451.0, 440.5] + 16	0.42
Aβ (µg/mL)	EC_50_ = 0.37 × NDSI_285_[728.0, 706.5] − 0.030	0.50	EC_50_ = 0.18 × RSI_285_[728.0, 706.5] − 0.15	0.52	EC_50_ = 0.020 × DSI_320_[741.0, 729.0] − 0.035	0.51

## Data Availability

Fluorescence spectral data are openly available in FigShare at https://doi.org/10.6084/m9.figshare.24265180, accessed on 19 October 2023.

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
