# Peer review of "On-Site Evaluation of Constituent Content and Functionality of Perilla frutescens var. crispa Using Fluorescence Spectra"

_molecules, 2023, doi:10.3390/molecules28207199_

Round 1
Reviewer 1 Report
Comments:
Koji Uwai and co-authors developed a facile method to analyze the activity and content of the components in leaves on-site based on the fluorescence spectra obtained by irradiating the perilla leaves with excitation light. This allows convenient, simultaneous, and non-destructive monitoring of the constituent contents and the functionalities of perilla leave and a simple harvest-time diagnosis. This work is interesting and detailed. After careful evaluation, the following issues are better to be addressed before it can be accepted.
1) For the accuracy and conciseness of the published works, it is better to re-organize the Figures 3-6, thus it will be readable.
2) For the wavelengths of the UV-vis, excitation and fluorescence emission, normally they are integers instead of decimals. While it showed the numbers of 718.5/715.5/433.5/583.5 in Table 1.
3) And section 4.4, the absorbance of the conditioned solution was measured at wavelengths 663.6, 646.6, and 750 nm. What are the exact meanings for the wavelengths of 663.6 and 646.6?
4) The Equation 4 should time 100% properly.
5) Please redraw the Table 2 with the split line, thus it will be readable.
Author Response
Dear Reviewer 1,
On behalf of my co-authors, we thank you very much for giving us an opportunity to revise our manuscript. We also appreciate you very much for their positive and constructive comments, and suggestions on our manuscript (molecules-2632666) entitled “On-site evaluation of constituent content and functionality of Perilla frutescens var. crispa using fluorescence spectra”. We have studied your comments carefully and have made revision in the resubmitted manuscript.
The detailed responses to the reviewers’ comments are listed below. We hope the revised manuscript is now suitable for publication in your journal.
Thank you for your consideration.
Yours sincerely,
October 7, 2023
Dr. Koji Uwai
E-mail: uwai@muroran-it.ac.jp

Reviewer 2 Report
Hidemichi Sano et al. evaluated the constituent contents, including Chlorophyl (Chl), total polyphenol content (TPC), total flavonoid content (TFC), rosmarinic acid (RA) and functional properties DPPH radical scavenging activity, ferric reducing antioxidant power (FRAP), oxygen radical absorbance capacity (ORAC), and Aβ aggregation inhibitory activity by using EEM. Three analysis methods were used, including NDSI, RSI, and DSI. R2 of 0.57 and 0.49 were achieved.
Overall, I appreciate the efforts the authors have made to measure so many contents and samples, but the analysis methods are too simple to achieve a decent regression result. Nowadays, with the development of machine learning, it is possible to get a much better result by using two-dimensional processing methods, e.g., CNN. And even CNN is not novel enough to publish, certain modification is required to optimize the prediction model.
While the methods are not novel, I encourage the authors to share the data, so that other teams from the world could use it to build a better model. Currently, it is stated by the authors that the data is not applicable, which weakens the contribution of the manuscript.
Author Response
Dear Reviewer 2,
On behalf of my co-authors, we thank you very much for giving us an opportunity to revise our manuscript. We also appreciate you very much for their positive and constructive comments, and suggestions on our manuscript (molecules-2632666) entitled “On-site evaluation of constituent content and functionality of Perilla frutescens var. crispa using fluorescence spectra”. We have studied reviewer’s comments carefully and have made revision in the resubmitted manuscript.
The detailed responses to the reviewers’ comments are listed below. We hope the revised manuscript is now suitable for publication in your journal.
Thank you for your consideration.
Yours sincerely,
October 7, 2023
Dr. Koji Uwai
E-mail: uwai@muroran-it.ac.jp

Reviewer 3 Report
The manuscript by Sano et al. proposes a correlation between the fluorescence emission profile of Perilla frutescens var crispa leaves with levels of antioxidants, romarinic acid and other bioactive markers .
The subject fits to the theme of the special issue in which has been submitted. The results part is quite clearly presented. Conversely discussion shows several weaknesses and limitations. First it seems it has not been properly revised since English fluency is quite poor and grammar mistakes clearly arising from lack of final proofreading are present. This obviously limits the stenght of the manuscript. Apart from style and grammar limitations, discussion section does not really focus on the discussion of achieved data and limitations of the proposed method but gives some limited comments.
In my opinion in order to have the manuscript accepted the authors should quite deeply revise the discussion section improving the style and grammar but also strengthening the discussion of the proposed method. I also would appreciate the authors to add a comment on the statistic relevance of using a 1 square cm of the leaves to assess whether the harvesting time is the optimal one.
Other points to be addressed are listed below:
Page 3, line 84 It should be: “Colors indicate the fluorescence intensity, from black (no fluorescence emission) to red (3000 a.u.)
In figure 2, I suggest to adjust the y axis of plot A to zero (currently it is -1.5). It will be also good to have Y-axis in plot B adjusted to zero (currently it starts from 100) in order to have all plots homogeneous
Plot H does not show any error bar. Was also the inhibitory activity towards amyloid aggregation assayed in triplicate?
In figure 2 label. It should be “standard error of the mean (SEM)”
Page 4, line 100 please delete the full stop after S35
In the discussion section, lines 1511-152 “Weak fluorescence at approximately 260–450 nm and 550–650 nm was observed with excitation below 350 nm (Figure 1) is the fluorescence spectrum of polyphenols covalently bound to the cell wall [30] whereas strong fluorescence at 650–800 nm was observed at 153 excitation wavelengths above 350 nm. is ascribed to chlorophyll emission from the chlo- 154 roplasts [29]…” It does not sound grammatically correct. Please revise
Experimental part, lines 231-232 the following sentence “At the same time, the rest was extracted with 99.5% ethanol (eth- 231 anol with a volume of five times the material weight) for three weeks” does not sound correct since an extraction lasting three weeks seems quite unusual.
Lines 258-259, I guess in A663.6 – 2.85×A646.6 A stands for absorbance (which should be mentioned) and numbers (weavelenghts) should be inserted ad lower cases. The same applies to formula (2)
Lines 264-265 “A 264 dilute solution of 80 µL of each extract” please rephrase
In section 4.7. RA quantification by LCMS details of mass spectrometer (ion source, mass analyzer) are missed. Furthermore, range of concentrations of rosmarinic acid used for the calibration are missed. Was any internal standard used for the analysis? Was SRM mode used?
Line 318 “which were defined as f0min” it is not clear
A general revision of English grammar and style is advisable.
As in my comments to the authors English style and grammar need to be improved
Author Response
Dear Reviewer 3,
On behalf of my co-authors, we thank you very much for giving us an opportunity to revise our manuscript. We also appreciate you very much for their positive and constructive comments, and suggestions on our manuscript (molecules-2632666) entitled “On-site evaluation of constituent content and functionality of Perilla frutescens var. crispa using fluorescence spectra”. We have studied reviewer’s comments carefully and have made revision in the resubmitted manuscript.
The detailed responses to the reviewers’ comments are listed below. We hope the revised manuscript is now suitable for publication in your journal.
Thank you for your consideration.
Yours sincerely,
October 7, 2023
Dr. Koji Uwai
E-mail: uwai@muroran-it.ac.jp

Round 2
Reviewer 2 Report
The data is shared by the authors now, which can be regarded as a contribution to the field. In this case, though the method is too direct, I would still agree for its acceptance.